# A Systems Approach Dissociates Fructose-Induced Liver Triglyceride from Hypertriglyceridemia and Hyperinsulinemia in Male Mice

**DOI:** 10.3390/nu13103642

**Published:** 2021-10-18

**Authors:** Ludivine Doridot, Sarah A. Hannou, Sarah A. Krawczyk, Wenxin Tong, Mi-Sung Kim, Gregory S. McElroy, Alan J. Fowler, Inna I. Astapova, Mark A. Herman

**Affiliations:** 1Division of Endocrinology, Beth Israel Deaconess Medical Center, Boston, MA 02215, USA; ludivine.doridot@inserm.fr (L.D.); sarahk.bethoney@gmail.com (S.A.K.); misung.kim1@gmail.com (M.-S.K.); gregory.mcelroy@northwestern.edu (G.S.M.); ajf115@georgetown.edu (A.J.F.); 2Duke Molecular Physiology Institute, Duke University, Durham, NC 27701, USA; sarah.hannou@duke.edu (S.A.H.); wenxin.tong@duke.edu (W.T.); inna.astapova@duke.edu (I.I.A.); 3Department of Pharmacology and Cancer Biology, Duke University, Durham, NC 27705, USA; 4Division of Endocrinology and Metabolism and Nutrition, Duke University, Durham, NC 27710, USA

**Keywords:** fructose, steatosis, hypertriglyceridemia, hyperinsulinemia, obesity, transcriptomics, ChREBP, SREPB1c, Tlr4

## Abstract

The metabolic syndrome (MetS), defined as the co-occurrence of disorders including obesity, dyslipidemia, insulin resistance, and hepatic steatosis, has become increasingly prevalent in the world over recent decades. Dietary and other environmental factors interacting with genetic predisposition are likely contributors to this epidemic. Among the involved dietary factors, excessive fructose consumption may be a key contributor. When fructose is consumed in large amounts, it can quickly produce many of the features of MetS both in humans and mice. The mechanisms by which fructose contributes to metabolic disease and its potential interactions with genetic factors in these processes remain uncertain. Here, we generated a small F2 genetic cohort of male mice derived from crossing fructose-sensitive and -resistant mouse strains to investigate the interrelationships between fructose-induced metabolic phenotypes and to identify hepatic transcriptional pathways that associate with these phenotypes. Our analysis indicates that the hepatic transcriptional pathways associated with fructose-induced hypertriglyceridemia and hyperinsulinemia are distinct from those that associate with fructose-mediated changes in body weight and liver triglyceride. These results suggest that multiple independent mechanisms and pathways may contribute to different aspects of fructose-induced metabolic disease.

## 1. Introduction

The metabolic syndrome (MetS) is a cluster of disorders including but not limited to obesity, dyslipidemia, hypertension, non-alcoholic fatty liver disease (NAFLD), hyperuricemia and insulin resistance which increase the risk of type 2 diabetes (T2D), cardiovascular disease (CVD), and some cancers [1,2,3]. The prevalence of MetS has increased markedly in the United States and worldwide over recent decades [4]. The causes of this epidemic are likely multifactorial with contributions from dietary and other environmental factors that interact with genetic predisposition. However, the precise contribution of diet, genetics, and other environmental factors to the prevalence of MetS overall, and to each of the distinct pathophysiological components of MetS remains unclear. Because of the pleiotropic effects of genes and environmental factors on metabolic traits and the difficulty in conducting long-term studies in which the dietary and environmental factors can be adequately controlled in human subjects, studies conducted in model organisms where both genetic background and environment can be precisely controlled will be essential to delineate the complex causal mechanisms underlying the diverse manifestations of this epidemic.

Increased consumption of sugar is one dietary factor that likely contributes to the burgeoning MetS epidemic [5]. Table sugar (sucrose) and high-fructose corn syrup constitute the major forms of sugar added to the food supply and are both comprised approximately equal parts glucose and fructose. Consumption of large amounts of sugar-sweetened beverages, a major source of added dietary sugar consistently associates with obesity, MetS, and risk for T2D and CVD in prospective human cohorts [5]. Excessive consumption of the fructose component of sugar appears to be particularly deleterious as high fructose diets can quickly cause or exacerbate many of the features of MetS in both humans and animal models [6,7]. Excessive fructose consumption may also be a major contributor to steatosis and may accelerate progression of NAFLD to more severe forms of liver disease [8].

The mechanism by which excessive fructose consumption can contribute to features of MetS is a topic of continued interest. Ketohexokinase (KHK) catalyzes the committed step in fructose metabolism, phosphorylating fructose to fructose-1-phosphate. Mice deficient for KHK have highly elevated blood and urine fructose levels when challenged with high-fructose diets, but are protected from fructose-induced metabolic disease [5]. This indicates that hyperfructosemia is not itself deleterious, and fructose metabolism is necessary for fructose-induced metabolic disease. Although recent data suggest that the intestine is a major site of fructose metabolism at low fructose doses, at higher doses, liver is a major site of fructose metabolism [9,10], and fructose metabolism within the liver is critical for the development of many aspects of fructose-induced metabolic disease [10,11,12]. Additionally, liver-specific knockout or knockdown of the fructose-responsive transcription factor carbohydrate responsive element binding protein (ChREBP, also known as Mlxipl) attenuates fructose-induced weight gain, hepatic de novo lipogenesis (DNL), VLDL secretion, and insulin resistance [13,14]. While knockdown of hepatic ChREBP attenuates fructose-induced hypertriglyceridemia and insulin resistance, this occurs without reductions in steatosis potentially due to its countervailing effects to reduce export of liver triglyceride in the form of VLDL [13,15]. These, and other genetic intervention studies suggest that there are likely multiple distinct molecular mechanisms and pathways that contribute to different aspects of fructose-induced metabolic disease.

We and others have observed marked variability in the susceptibility of distinct mouse strains to metabolic disease depending on dietary macronutrient content. For instance, Surwit et al. noted that C57Bl/6J mice, but not A/J mice, develop obesity and diabetes in response to high-fat, but not high-sucrose diets [16,17]. Likewise, Nagata et al. showed that C3H and CBA strains are prone to fructose-induced hypertriglyceridemia and hyperinsulinemia, and C57BL/6N and DBA strains are resistant [18]. Based on these observations, we hypothesized that a genetic cross between fructose-sensitive and -resistant strains might help illuminate mechanistic relationships between fructose-induced dysmetabolic features. We therefore generated a small pilot F2 cohort by intercrossing the fructose-sensitive (C3H/HeJ) and fructose-resistant (C57BL/6J) strain to determine whether a systems approach may be tractable for defining the pathogenesis of fructose-induced disease. Our results presented here demonstrate that within this well-controlled cohort, fructose-induced weight gain and liver triglyceride are independent of fructose-induced hypertriglyceridemia and hyperinsulinemia. These results indicate that additional studies in genetic crosses of fructose-sensitive and -resistant strains will be useful for identifying genes and delineating distinct mechanistic pathways by which fructose consumption impacts diverse metabolic traits.

## 2. Materials and Methods

### 2.1. Animals and Diets

All studies were approved by Beth Israel Deaconess Medical Center Institutional Animal Care and Research Advisory Committee. Both male and female mice were studied at ages as indicated in specific experiments. C57BL/6J (Stock# 000664) and C3H/HeJ (Stock# 000659) mice were obtained from The Jackson Laboratory. Mice were housed and bred in a temperature-controlled animal facility with a 12-h light/12-hr dark cycle and free access to food and water at Beth Israel Deaconess Medical Center (Boston, MA, USA). Before initiation of diet studies, the mice were maintained on a standard laboratory chow diet (Purina lab Diet 5008). The 60% fructose diet (HFrD, TD.89247, Harlan Teklad) contained 20.2% protein, 12.9% fat, and 66.8% carbohydrate.

### 2.2. Body Weight and Metabolic Parameters

Body weight was measured on a weekly basis. The F2 cohort was euthanized after 4 weeks of ad libitum HFrD. Analyses were performed using blood at euthanasia after an overnight fast followed by 3 h refeeding. Serum glucose was measured using glucose oxidase reagent (Thermo Scientific, USA). Serum insulin was measured by ELISA (Crystal Chem Ultra Sensitive Mouse Insulin ELISA). Serum FGF21 was measured by ELISA (R&D Systems, Mouse/Rat FGF-21 Quantikine ELISA). Serum and liver triglyceride and cholesterol levels were measured by colorimetric assays (Stanbio Laboratory, USA). Uric acid was measured by colorimetric assay (Stanbio Laboratory, USA).

### 2.3. Liver Triglyceride and Cholesterol Measurements

Neutral lipids were extracted from a piece of pre-weighed liver using the Folch method [19]. Briefly liver was homogenized with chloroform/methanol (2:1) and the two phases were separated with the addition of 0.2× volumes of 0.9% NaCl. A portion of the triglyceride containing chloroform phase was collected, evaporated, and re-suspended in a mixture of 2:1 [Butantol: (3:1 Triton: MeOH)]. Triglyceride and cholesterol were then measured using kits purchased from StanBio according to the manufacturer’s protocol and normalized to liver weight.

### 2.4. RNA Extraction and Quantitative PCR

TRI reagent (MRC, catalog no. TR118) was used for RNA isolation from mouse liver and adipose tissue. RNA was reverse transcribed using SuperScript VILO kit (ThermoFisher Scientific, USA). Gene expression was analyzed with the ABI Prism sequence detection system (SYBR Green; ThermoFisher Scientific, USA). Gene-specific primers were synthesized by IDT (Integrated DNA technology) (see Date 1 and Data 2 in the Appendix A). Each sample was run in duplicate and normalized to Rplp0 mRNA.

### 2.5. SCRB-Seq, Gene Network Analysis, and Gene Set Enrichment Analysis

Due to practical constraints related to sample processing for the plate-based SCRB-seq method, only 48 of the 53 male mice were used in this analysis. Five male mice with serum insulin and triglyceride levels in closest proximity to the median for the cohort were excluded at this step. RNA was prepared from livers of individual animals as described above and was treated with DNAse (Turbo DNAse, Ambion, USA) then equilibrated at 50 ng/uL. SCRB-seq sequencing, read alignment, and generation of raw gene counts were conducted by the Broad Institute Genomics Core as described in [20]. The raw counts for each gene in each sample were then normalized using EdgeR [21]. Unsupervised clustering demonstrated that one sample segregated apart from all other samples suggestive of contamination or degradation. This sample was removed from the analysis. Thus, the transcriptomic analysis was conducted with 47 male F2 mice.

Gene network analysis was performed using the Gene Whole co-Expression Network Analysis (GWENA) package available through Bioconductor [22]. Genes with fewer than 10 reads in at least one sample were excluded from the analysis. Of the remaining genes, 50% of the most variable genes were retained for further analysis.

Gene modules generated using GWENA were generated as “unsigned” modules to enhance power for trait association. To conduct gene set enrichment analysis, principal component analysis was performed on the set of genes included in each module that associated significantly with one or more metabolic trait. The weights associated with the first principal component for each module were assigned to the associated gene in that module in order to segregate genes that are positive versus negative contributors to the module. Gene set enrichment analysis was conducted using EnrichR separately for gene sets composed of positively and negatively associated genes for each module. Databases used in this analysis included “GO Biological Process 2021” and “TRRUST Transcription Factors 2019” [23,24]. Data provided in the data supplement includes the top 20 sets ranked by *p*-value for each analysis of the relevant modules.

### 2.6. SNP Genotyping

Taqman SNP Genotyping assays (ThermoFisher Scientific, USA) for allelic discrimination were generated for SREBF1 and TLR4 for rs26973133 and rs3023006, respectively. Assays were performed on an ABI Prism sequence detection system (ThermoFisher Scientific, USA) or a QuantStudio 6 Flex instrument (ThermoFisher Scientific, USA).

### 2.7. Statistical Analysis

Results are presented as the mean ± S.E unless otherwise noted in the text or figure legend. Differences between groups were examined for statistical significance by either Student’s *t*-test or two-way ANOVA followed by Tukey’s post hoc test as appropriate using Prism statistical software. Metabolic trait correlations and the Fligner-Killeen tests of homogeneity of variance were assessed using R 4.0 statistical software as was SNP genotyping which was assessed by linear regression.

## 3. Results

### 3.1. C3H and C57 Mice Are Sensitive and Resistant, Respectively to Fructose-Induced Hyperinsulinemia and Hypertriglyceridemia

Eight-week-old, male C3H/HeJ (C3H) and C57BL/6J (C57) were subjected to ad libitum chow or high-fructose diet (HFrD) for 6 weeks. C3H mice tended to weigh more after HFrD feeding compared to chow (HFrD: 4.4 ± 0.33 g vs. chow: 2.9 ± 0.42, *p* = 0.07) and weighed significantly more than C57 mice fed HFrD at the end of 6 weeks (Table 1). C57 mice on HFrD did not gain weight over this 6-week feeding period. Serum metabolic parameters were assessed after an overnight fast followed by 3 h refeeding. Refed serum glucose did not differ between strains or diet (Table 1). In chow fed mice, serum insulin was 50% lower in C3H compared to C57 mice. However, HFrD produced a 2.3-fold increase in serum insulin in C3H mice compared to chow-fed C3H whereas no increase in serum insulin occurred in HFrD-C57 mice. There was no difference in serum triglyceride levels between C3H and C57 mice on chow diets. HFrD increased circulating triglycerides in C3H mice, but had no effect on serum triglycerides in C57 mice (Table 1). Serum cholesterol levels tended to be higher in C3H mice compared to C57 mice and tended to increase with HFrD in both strains, but these differences were not statistically significant. Hepatic triglyceride levels tended to be higher in chow-fed C3H compared to chow-fed C57 and increased ~3-fold with HFrD in both strains. Hepatic cholesterol levels did not differ between strains or diet. These results are comparable to those previously observed by Nagata et al. [18].

As hepatic fructose metabolism and ChREBP activity are essential for aspects of fructose-induced disease, we examined expression of ChREBP isoforms and selected hepatic ChREBP transcriptional targets involved in fructose metabolism and other key metabolic processes such as DNL, glycolysis, and glucose production. Fructose feeding increased the expression of the potent ChREBP-β isoform in both C57 and C3H mice and this increase was more substantial in C3H compared to C57 (Figure 1). The increase in ChREBP-β with HFrD in C3H compared to C57 mice was paralleled by more robust increases in genes involved in fructose metabolism including Slc2a5 (Glut5, a fructose transporter), ketohexokinase (KHK), aldolase b (Aldob), and triokinase and FMN cyclase (Tkfc), as well as DNL enzymes including fatty acid synthase (Fasn) and acetyl-CoA carboxylase (Acaca) in C3H compared to C57 mice (Figure 1). Differences in the response of enzymes involved in glucose metabolism were less distinct between the two strains. The differences in expression of DNL and fructolytic enzyme induction between C3H and C57 mice could potentially contribute to the differential susceptibility to fructose-induced disease.

### 3.2. An F2 Cohort Demonstrates Larger Variance in Most Fructose-Affected Metabolic Traits Compared to the Parental Strains

Following confirmation of strain-dependent susceptibilities to fructose-induced metabolic traits and hepatic gene expression, we generated a small F2 cohort of 98 mice (45 females and 53 males) derived from intercrossing C3H and C57 F0 mice. At weaning, these mice were started on HFrD and maintained on this diet for 4 weeks. As the median and variance in fructose-induced hyperinsulinemia and hypertriglyceridemia were markedly higher in male compared to female mice (Figure 2A), we focused on male mice for subsequent analyses including body weight, serum insulin, glucose, triglycerides, and cholesterol, and liver triglyceride and cholesterol levels (Figure 2B). Male F2 mice displayed larger variance in key metabolic traits including serum insulin, glucose, and triglycerides, and both the median and variance appeared markedly higher in the F2 cohort compared to either parental strain for serum glucose or triglycerides (Figure 2B). Serum cholesterol in the F2 cohort spans the range defined by the parental strains. In contrast, body weight and hepatic triglycerides tend to be higher in the C3H parental strain compared to the F2 cohort. The median hepatic cholesterol was lower in F2 mice than either parental strain. It should be noted that the animals in the parental strain comparison groups were euthanized at 14 weeks of age whereas the F2 mice were euthanized at 8 weeks of age which may contribute to some of these differences. The larger variance in many of the serum metabolic parameters in the F2 cohort compared to parental strains suggests complex polygenic contributions to these fructose-induced metabolic traits as is commonly the case for other complex mouse and human metabolic traits.

We next examined the relationship between different metabolic traits among mice within this fructose-fed F2 cohort. HFrD can quickly induce weight gain, and weight gain is often associated with other metabolic disturbances like insulin resistance and hypertriglyceridemia. Consistent with this, body weight correlated with serum insulin (R = 0.45, *p* < 0.01) and glycemia (R = 0.4, *p* < 0.01) (Table 2). However, correlations between body weight and serum or hepatic triglyceride or cholesterol levels did not achieve statistical significance. Although serum insulin levels correlate with body weight and serum triglyceride levels do not, serum insulin and triglyceride levels correlate with each other (R = 0.44, *p* < 0.01). This suggests that the mechanisms that mediate the association between body weight and hyperinsulinemia may be in part distinct from the mechanisms that mediate the association between hyperinsulinemia and hypertriglyceridemia within this cohort. Although it is commonly hypothesized that liver triglyceride contributes to hypertriglyceridemia and systemic insulin resistance, in this cohort, there was no significant correlation between hepatic triglyceride levels and either serum triglyceride or serum insulin, indicating that liver fat per se was not a major determinant of either fructose-induced hypertriglyceridemia or hyperinsulinemia. However, hepatic triglyceride levels did weakly correlate with serum glucose (R = 0.32, *p* < 0.05) suggesting that hepatic lipid levels may impact glycemia (or vice versa) independently of serum insulin which is commonly considered an index of insulin resistance.

FGF21 is a metabolic hormone produced by the liver that can be induced by fructose consumption in a ChREBP-dependent manner [25,26,27,28,29]. In humans and animal models, levels of circulating FGF21 associate with increased cardiometabolic risk factors including obesity and dyslipidemia [30]. Despite this association, FGF21 has pleiotropic metabolic actions to enhance metabolism including effects to reduce body weight, liver triglyceride, enhance systemic glucose, and lipid metabolism, and reduce sugar consumption [30]. In this cohort, circulating FGF21 did not associate with body weight, correlated positively with serum insulin, serum triglyceride, and serum cholesterol levels, and correlated inversely with liver fat content (Table 2). Uric acid is produced in association with fructose metabolism as the robust phosphorylation of fructose by KHK depletes free phosphate which stimulates purine degradation and uric acid production [5]. Uric acid produced as a result of fructose metabolism has been postulated to contribute to the pathogenesis of metabolic disease. In this cohort, we observed that serum urate correlates positively with serum triglycerides, but not with any other measured metabolic traits.

### 3.3. A Variant in the Srebp1 Locus Does Not Associate with Metabolic Phenotype in the F2 Cohort

Nagata et al. previously proposed that differences in fructose susceptibility between different mouse strains may be due to a single nucleotide polymorphism in the promoter of the Srebf1 gene, which encodes Srebp1c, a transcription factor that is responsive to fructose consumption and that regulates metabolic programs including lipogenesis [18,31,32]. To determine if this polymorphism contributed to metabolic phenotypes in this cohort, we performed targeted SNP genotyping of the F2 cohort for rs26973133 (chr11:60210632-60211132, mouse genome assembly mm10) which is located in the 5′ untranslated region of the Srebf1 gene and discriminates the C3H and C57 haplotypes in this region [33]. We were unable to detect significant associations between Srebpf1 genotype and any measured metabolic trait (Figure 3), indicating that differences in these Srebf1 haplotypes are not a major determinant of the susceptibility to fructose-induced dysmetabolism among C3H and C57 mice.

### 3.4. The Inactivating Missense Variant in Tlr4 Present in C3H/HeJ Mice Does Not Associate with Metabolic Phenotype in the F2 Cohort

Recent studies suggested that fructose induces gut dysbiosis and impairs intestinal barrier function leading to portal endotoxemia which can precipitate hepatic inflammation contributing to fructose-induced derangements in lipid homeostasis and metabolic disease [34,35,36]. C3H/HeJ mice harbor a natural inactivating mutation in the Tlr4 gene which is the lipopolysaccharide (LPS) receptor [37,38]. This renders them resistant to the effects of LPS-induced inflammation. Our results indicate that C3H mice are susceptible to fructose-induced disease whereas C57 mice are not, which implies that the adverse effects of lipopolysaccharide mediated through Tlr4 are not necessary for fructose-induced disease (Table 1). To further examine whether Tlr4 can impact fructose-induced dysmetabolism, we assessed Tlr4 haplotypes in the F2 cohort (rs3023006, chr4:66840856-66841356, mouse genome assembly mm10). Tlr4 genotype showed no association with any measured metabolic trait indicating that LPS-mediated signaling and inflammation is not a major determinant of fructose-mediated metabolic effects in this cohort (Figure 4).

### 3.5. Distinct Hepatic Gene Programs Associate with Fructose-Induced Hypertriglyceridemia and Hyperinsulinemia Compared to Hepatic Triglyceride Levels

As metabolism of fructose in the liver is critical for fructose-induced metabolic disease and hepatic fructose metabolism is under the control of nutrient responsive transcription factors, we sought to define hepatic transcriptional networks underlying fructose-induced metabolic disease in this cohort. We measured gene expression by performing SCRB-Seq on liver tissue samples from the 48 male mice from the F2 cohort [20]. We analyzed hepatic transcriptomic data using a modified version of weighted correlation network analysis after filtering out genes with low expression levels and focused on a set of 3962 most highly variably expressed genes to identify co-regulated gene sets within the cohort [22,39]. This analysis revealed 6 distinct sets of co-regulated genes (gene modules). An “eigengene” was computed for each module. This is defined in GWENA/WGCNA analysis by the first principal component of the expression levels of the genes included in a module and can be interpreted as the “expression” level of the module for each individual animal in the cohort. We regressed the “eigengenes” derived from these modules against measured phenotypes within the F2 cohort. Although body weight did not correlate with liver triglyceride and liver cholesterol within the cohort as a whole, Module 2 associated inversely with body weight, hepatic triglycerides, and hepatic cholesterol levels (Figure 5). Module 3 associated positively with hepatic triglyceride, negatively with serum triglyceride and serum insulin, and showed no correlation with body weight or other measured traits. Module 5 associated positively with hepatic triglyceride and hepatic cholesterol levels, and inversely with serum cholesterol levels. Modules 0, 1, and 4 did not associate significantly with any measured trait.

### 3.6. Gene Set Enrichment Analysis

To infer transcription factors that might contribute to the regulation of distinct hepatic gene modules, their biological function, and their associated metabolic traits, we performed gene set enrichment analysis on relevant hepatic gene modules for gene ontology pathways and transcription factors using the EnrichR platform (Table 3, Appendix A) [40]. Positively weighted genes in Module 2 that correlated inversely with body weight, hepatic triglyceride, and hepatic cholesterol associated strongly with gene ontology sets involved in glycosylation of proteins and the function of the endoplasmic reticulum such as protein N-linked glycosylation via asparagine (GO:0018279), protein exit from endoplasmic reticulum (GO:0032527), response to unfolded protein (GO:0006986), response to endoplasmic reticulum stress (GO:0034976), and ubiquitin-dependent ERAD pathway (GO:0030433) among others. Positively weighted genes in Module 2 were enriched for genes associated with transcription factors implicated in ER stress and ER function including ATF6 and XBP1 as well as PPARγ. Negatively weighted genes in Module 2 were enriched for biological processes including cholesterol efflux (GO:0033344), high-density lipoprotein particle remodeling (GO:0034375), cholesterol homeostasis (GO:0042632), and lipid transport (GO:0006869). Negatively weighted genes were enriched for known targets of HNF4A.

Positively weighted genes in Module 3, which correlated positively with liver triglyceride and negatively with serum insulin and serum triglyceride were enriched for sets associated with fatty acid oxidation such as fatty acid beta-oxidation (GO:0006635) and highly enriched for PPARα gene targets. Negatively weighted genes in Module 3 were enriched for sets associated with lipid synthesis including regulation of lipid metabolic process (GO:0019216), cholesterol biosynthetic process (GO:0006695), lipid biosynthetic process (GO:0008610), and triglyceride biosynthetic process (GO:0019432). These negatively weighted genes were enriched for known targets of the lipogenic transcription factors Srebf1 and Mlxipl (ChREBP). Negatively weighted genes in this module were also enriched for targets of Tfap2a (also known as activating enhancer-binding protein 2-alpha) which is also implicated in regulation of lipid droplets, as well as the transcription factor Sp1 which can regulate and function cooperatively with Srebp1c [41,42,43].

Gene ontology and transcription factor associations with genes in Module 5 were less robust than for Modules 2 and 3 with few significant associations (Appendix A).

## 4. Discussion

Excessive fructose consumption can rapidly induce or exacerbate features of MetS in humans and rodent models and recent work indicates hepatic fructose metabolism is essential for fructose-induced disease. The mechanisms by which fructose contributes to metabolic disease remain incompletely understood. However, the rapid metabolism of fructose in the liver may contribute to the pathogenesis of metabolic disease through its effects to cause energetic stress, induce uric acid production, activate gene programs to enhance lipogenesis, and/or provide substrate that might be used in lipogenesis or other anabolic processes. Fructose may provide lipogenic substrate both via its metabolism in the liver or indirectly through microbial production of acetate which is then used in the liver [44]. In either case, fructose-induced lipogenesis may contribute to liver fat and as liver fat content commonly associates with hypertriglyceridemia and insulin resistance in human populations, mechanisms that increase liver fat or otherwise cause derangements in hepatic lipid metabolism are often invoked as causal mediators of insulin resistance although this remains controversial [45,46].

We present, for the first time, an examination of fructose-mediated metabolic traits in a genetic cross between fructose-sensitive and -resistant mouse strains. By performing transcriptomic analysis in the livers of this genetically heterogenous cohort, we can determine in an unbiased manner, which traits associate with which metabolic programs. Nutrients may contribute to the development of obesity and metabolic disease in part by regulating gene expression programs, like those controlled by ChREBP and SREBP1c [15,47,48]. Distinct gene programs and pathways may interact to regulate the various related metabolic traits associated with MetS. Using a low-cost, high-throughput sequencing method (SCRB-seq), we quantified the abundance of ~4000 highly variable mRNA transcripts in liver of male F2 mice. We defined sets of co-regulated gene modules using whole-genome correlation network analysis and demonstrated that distinct sets of genes associate with different combinations of metabolic traits. In general, these results suggest that the superposition of quasi-independent mechanistic pathways will in aggregate determine the level of distinct traits in individual animals. This analysis can facilitate examination of causal linkages between different traits. For instance, while body weight correlates with hyperinsulinemia within the cohort overall, the mechanistic program defined by Module 3 strongly associates with hyperinsulinemia and circulating and liver triglycerides but shows no association with body weight. Thus, a major mechanism driving the intercorrelation between circulating insulin and triglyceride is largely independent of the body weight in this fructose-fed cohort. This suggests that the dyslipidemia in human populations associated with consumption of sugar-sweetened beverages (SSBs) may be in part independent of SSB-associated weight gain [5].

Importantly, within this cohort, we demonstrate that fructose-induced liver triglyceride is independent of fructose-induced hyperinsulinemia and hypertriglyceridemia. This result may seem surprising since fructose is known to induce both lipogenesis and increased circulating lipids and since liver fat content and hypertriglyceridemia are so often observed together in human populations. Nevertheless, these results suggest that the processes by which fructose may enhance liver triglyceride and circulating lipids are largely independent of each other. In humans and animal models, increased DNL is thought to be a significant contributor to the development of steatosis [49,50]. The increase in DNL in obesity and NAFLD associates with increased expression of enzymes of fatty acid synthesis. As fructose potently enhances expression of the enzymes of fatty acid synthesis and can provide substrate for DNL, one might expect that coordinate induction of the enzymes involved in fatty acid synthesis as observed in Module 3 would associate with increased liver fat. In this genetically heterogenous cohort, the hypothesis is incorrect. Rather, the enzymes of fatty acid synthesis associate with circulating triglyceride levels and inversely with liver triglyceride levels. The enzymes of fatty acid synthesis are also co-regulated with enzymes involved in VLDL packaging and export, and these latter enzymes may increase circulating triglycerides and decrease liver triglycerides. Our results suggest the effects of VLDL packaging and export may dominate over the effects of new fatty acid synthesis resulting in a net decrease in liver fat. Moreover, this indicates that the predominant effects fructose-mediated activation of ChREBP and SREBP1c may be to enhance circulating triglycerides rather than liver fat content. This is concordant with the results from genetic interventions targeting ChREBP in animals as well as genetic variants in the ChREBP locus that impact circulating triglycerides in human populations [13,15,51,52,53].

One limitation of this study is that we cannot infer the direction of causality between the observed co-regulated genes and associated metabolic traits. Nevertheless, gene set enrichment analysis suggests a number of interesting mechanistic hypotheses. For instance, Module 3 was enriched for ChREBP and SREPB1c transcriptional targets including enzymes involved in DNL and strongly associated with hyperinsulinemia and hypertriglyceridemia, and inversely with liver triglyceride. Insulin activates hepatic SREBP1c activity which is synergistic with ChREBP activity on lipogenic gene expression [15,32]. ChREBP enhances VLDL secretion in part through regulation of MTTP [15,48], and variants in the ChREBP locus are associated with hypertriglyceridemia in humans [51,52,53]. Moreover, hepatic ChREBP activity is essential for fructose-induced hyperinsulinemia and insulin resistance [13,14]. Thus, it is possible that the association between this module 3 and hyperinsulinemia and hypertriglyceridemia is bidirectional. Importantly, whatever the causal mechanism, Module 3 inversely associated with liver triglyceride. Again, this indicates that the mechanisms inducing hypertriglyceridemia and hyperinsulinemia in association with this gene program are not caused by increased liver fat content.

As previously noted, we and others have reported that fructose feeding markedly upregulates expression of the metabolic hormone FGF21 in a ChREBP-dependent manner. However, within this cohort, hepatic FGF21 expression and circulating FGF21 did not appear to correlate strongly with other ChREBP transcriptional targets that are co-regulated in Module 3. Thus, while ChREBP is capable of inducing FGF21, and ChREBP is necessary for fructose-mediated induction of FGF21, ChREBP may not be a major contributor to the variance in hepatic expression and circulating levels of FGF21 in the setting of chronic fructose feeding or other obesogenic states.

This study has several additional limitations. The study is largely observational and hypothesis generating. Without anchoring phenotypes or gene expression programs in genetic variation, we are unable to determine causality. This cohort may be too small for robust genetic analysis unless there are variants with very large effect sizes. Nevertheless, this study suggests that a larger, fructose fed F2 cohort derived from these strains might provide a tractable approach to identify the genetic variants that causally contribute to fructose-dependent metabolic traits and gene expression programs. Even so, the results presented here suggest interesting hypotheses that can be pursued further by targeted genetic interventions in animal models. Moreover, we can use this cohort to test candidate genes previously implicated in the pathogenesis of fructose-induced disease. A prior study had suggested that variants proximal to the Srebf1 gene impacted susceptibility to fructose-induced disease [18]. However, our data demonstrate that C57 versus C3H haplotypes in the Srebf1 locus have no detectable impact on metabolic traits in this fructose-fed cohort. Similarly, recent work suggests that Tlr4 is an important mediator of fructose-induced signaling on hepatic lipogenesis and liver triglyceride [36]. In contrast, our results show that an inactivating mutation in Tlr4 had no impact on lipid phenotypes in this cohort. This does not exclude the possibility that inflammation might impact metabolic phenotypes through Tlr4 with prolonged fructose exposure for more than 30 weeks as suggested by Todoric et al. [36]. Nevertheless, these results demonstrate that Tlr4 signaling is not essential for fructose-mediated dysmetabolism.

Another limitation of this study is that molecular analyses were limited to the liver in this cohort. While the liver plays a critical role in fructose-induced metabolic disease, other tissues like the adipose tissue are also major regulators of systemic fuel homeostasis. Moreover, since the generation of this cohort, we and others have identified an important role for the intestine in fructose metabolism and that was not assessed here [10,11,13,44]. Metabolic phenotyping was also limited. Mice were fed high-fructose diets ad libitum. Food intake, energy expenditure, and body composition were not measured. Thus, genetic variation having large effects on feeding behavior or thermogenesis might have large impacts on the phenotypes assessed here, but that was not directly assessed. However, as noted above, body weight within this cohort had little impact on important metabolic traits including serum and hepatic triglyceride and cholesterol levels. This study will provide motivation for larger cohorts with analysis of more tissues, combined with genetic analysis in a true systems genetics approach to decipher mechanisms contributing to fructose-induced metabolic disease.

## Figures and Tables

**Figure 1 nutrients-13-03642-f001:**
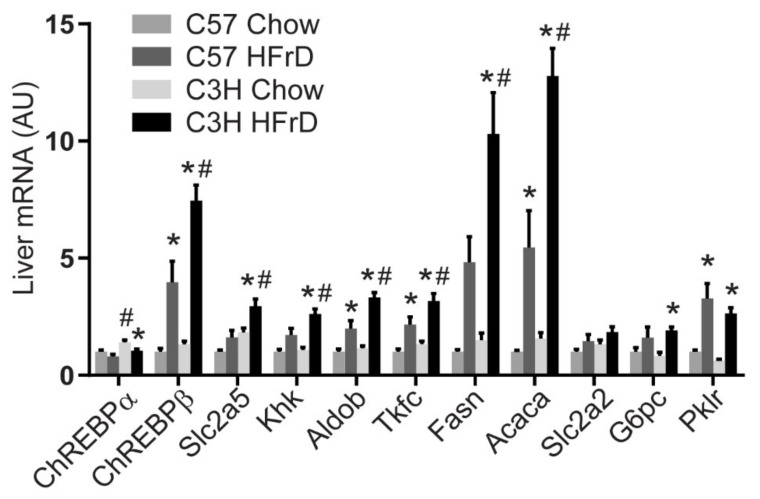
mRNA expression of genes involved in fructose and glucose metabolism in the liver of male C57BL/6J and C3H/HeJ mice fed standard chow or HFrD for 6 weeks. Mice were euthanized following an overnight fast and 3 h refeed. * *p* < 0.05 comparing diets within strain; # *p* < 0.05 comparing strains within diet. n = 6/group.

**Figure 2 nutrients-13-03642-f002:**
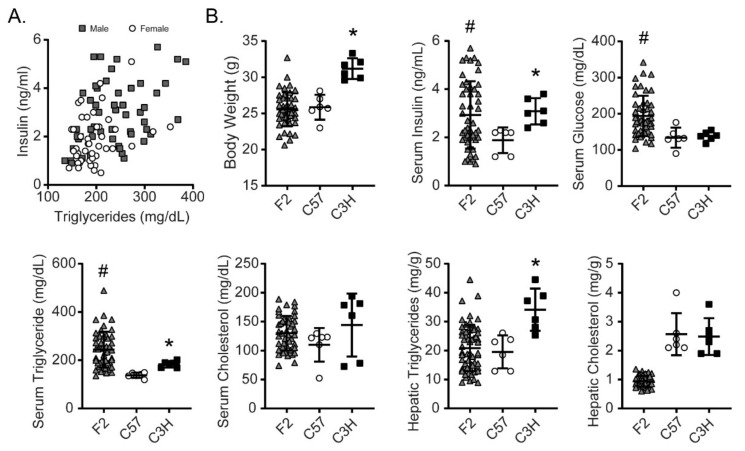
(**A**) Comparison of the distributions of serum insulin and triglyceride levels in male and female F2 mice after 4 weeks high-fructose feeding. (**B**) Distribution of metabolic traits within the male F2 mice compared to male parental C57 and C3H mice after high-fructose feeding. * *p* < 0.05 comparing means of C57 and C3H as in Table 1. # *p* < 0.05 for inhomogeneity of variance.

**Figure 3 nutrients-13-03642-f003:**
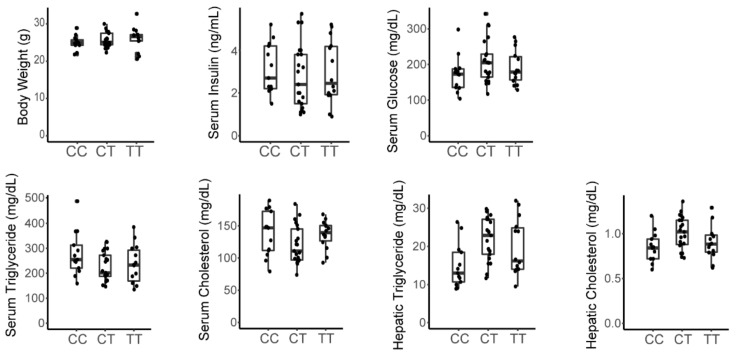
Srebf1 haplotypes do not associate with metabolic traits in male fructose fed F2 mice. Genotyping performed for rs26973133 [C/T]. CC = homozygous C3H; CT = heterozygous; TT = homozygous C57.

**Figure 4 nutrients-13-03642-f004:**
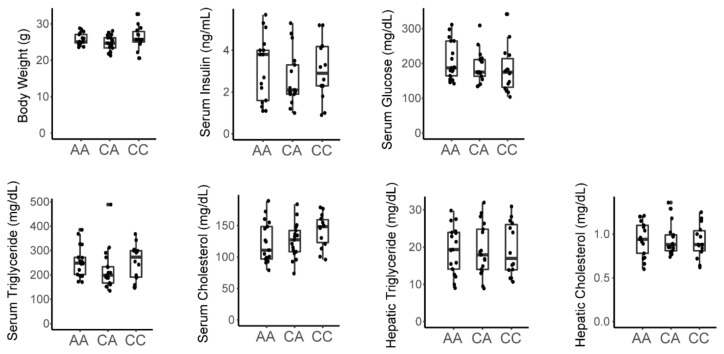
An inactivating missense variant in Tlr4 does not associate with metabolic traits in male fructose fed F2 mice. Genotyping performed at rs3023006 [C/A]. AA = homozygous C3H; CA = heterozygous; CC = homozygous C57.

**Figure 5 nutrients-13-03642-f005:**
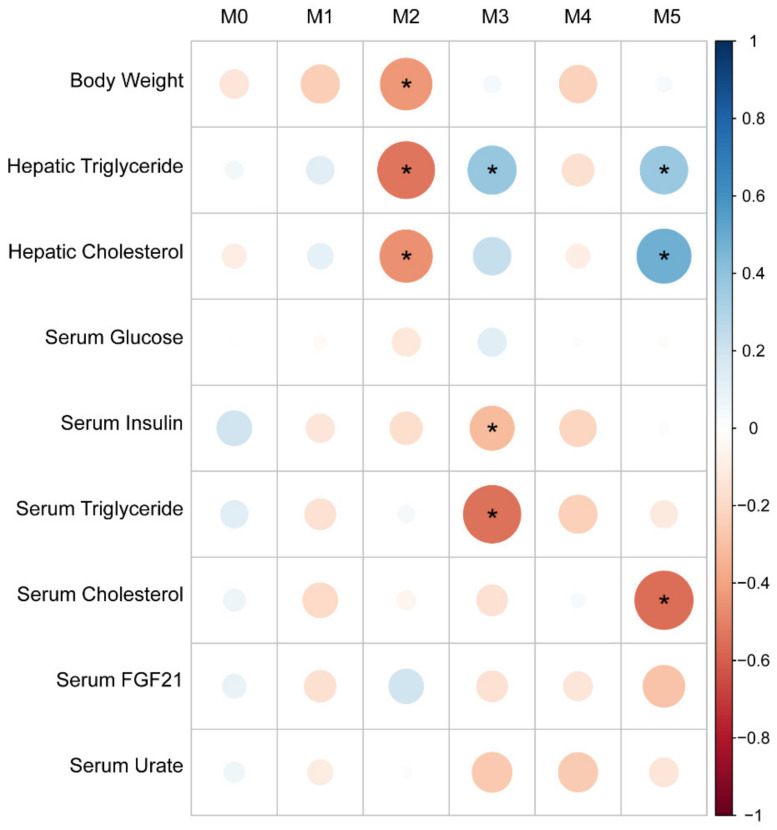
Gene modules (M0—M5) associate with distinct combinations of metabolic traits in male F2 mice. Size and color correspond to the correlation coefficient. * *p*-value < 0.05.

**Table 1 nutrients-13-03642-t001:** Metabolic phenotype in male C57BL/6J versus C3H/HeJ mice fed chow versus high-fructose diet for 6 weeks.

	C57BL/6J-Chow	C57BL/6J-Fructose	C3H/HeJ-Chow	C3H/HeJ-Fructose
Body Weight (g) Body Weight Change (g)	28 ± 0.7 0.6 ± 0.3	25.6 ± 0.6 −1.1 ± 0.5 *	29.2 ± 1.0 2.9 ± 0.4 #	32.2 ± 0.6 # 4.4 ± 0.3 *,#
Serum Glucose (mg/dL)	154 ± 11	134 ± 11	129 ± 11	138 ± 5
Serum Insulin (ng/mL)	2.5 ± 0.3	1.9 ± 0.2	1.3 ± 0.2	3.1 ± 0.2 *,#
Serum Triglyceride (mg/dL)	137 ± 5	138 ± 5	149 ± 9	183 ± 5 *,#
Serum Cholesterol (mg/dL)	93 ± 7	110 ± 12	129 ± 9	144 ± 22
Hepatic Triglyceride (mg/g)	6.2 ± 0.7	19.5 ± 2.3 *	10.1 ± 1.5	34.1 ± 3.0 *,#
Hepatic Cholesterol (mg/g)	2.1 ± 0.1	2.6 ± 0.3	2.4 ± 0.1	2.5 ± 0.3

* *p* < 0.05 comparing diet within genotype; # *p* < 0.05 comparing genotypes within diet.

**Table 2 nutrients-13-03642-t002:** Pearson intercorrelations (R) between metabolic traits within male F2 cohort.

	Body Weight	Serum Insulin	Serum Glucose	Serum Trig.	Serum Chol.	Hepatic Trig.	Hepatic Chol.	Serum FGF21
Body Weight	—							
Serum Insulin	0.45 **	—						
Serum Glucose	0.40 **	0.26	—					
Serum Triglyceride	0.27	0.44 **	−0.12	—				
Serum Cholesterol	0.26	0.04	0.06	0.29 *	—			
Hepatic Triglyceride	0.20	0.08	0.32 *	−0.21	−0.23	—		
Hepatic Cholesterol	0.08	0.08	0.29 *	−0.19	−0.24	0.68 ****	—	
Serum FGF21	−0.04	0.29 *	−0.25	0.43 **	0.32 *	−0.33 *	−0.24	—
Serum Urate	0.23	0.12	−0.11	0.46 ***	0.19	−0.15	−0.14	−0.01

*p*-value * <0.05, ** <0.01, *** <0.001, **** <0.0001.

**Table 3 nutrients-13-03642-t003:** Top gene sets ranked by *p*-value for modules 2 and 3 using EnrichR gene set enrichment analysis and the Gene Ontology Bio Process and TRRUST databases.

Module 2	Overlap	−log10 (*p*-Value)
** *Gene Ontology Bio Process: Upregulated* **		
protein N-linked glycosylation via asparagine (GO:0018279)	12/30	13.0
peptidyl-asparagine modification (GO:0018196)	12/31	12.8
protein exit from endoplasmic reticulum (GO:0032527)	11/24	12.7
response to unfolded protein (GO:0006986)	14/49	12.7
response to endoplasmic reticulum stress (GO:0034976)	19/110	12.7
** *Gene Ontology Bio Process: Down Regulated* **		
cholesterol efflux (GO:0033344)	6/24	6.5
high-density lipoprotein particle remodeling (GO:0034375)	5/18	5.8
cholesterol homeostasis (GO:0042632)	8/71	5.7
sterol homeostasis (GO:0055092)	8/72	5.7
cholesterol transport (GO:0030301)	7/51	5.7
** *TRRUST 2019: Upregulated* **		
ATF6 human	3/14	2.7
XBP1 human	3/19	2.3
HSF1 human	3/31	1.7
PPARG human	4/66	1.4
HDAC9 human	2/18	1.4
MYC mouse	3/49	1.2
** *TRRUST 2019: Downregulated* **		
HNF4A mouse	5/36	4.2
HNF4A human	5/45	3.7
NR1H4 human	3/20	2.8
ESRRA mouse	2/8	2.4
NR1H4 mouse	2/11	2.1
RORA mouse	2/11	2.1
**Module 3**	**Overlap**	**−log10 (*p*-Value)**
** *Gene Ontology Bio Process: Upregulated* **		
fatty acid beta-oxidation (GO:0006635)	19/52	25.8
fatty acid catabolic process (GO:0009062)	18/70	21.3
fatty acid oxidation (GO:0019395)	16/59	19.4
fatty acid beta-oxidation using acyl-CoA oxidase (GO:0033540)	9/15	15.0
long-chain fatty acid metabolic process (GO:0001676)	14/83	13.9
** *Gene Ontology Bio Process: Down Regulated* **		
regulation of lipid metabolic process (GO:0019216)	15/92	16.6
cholesterol biosynthetic process (GO:0006695)	10/35	13.9
sterol biosynthetic process (GO:0016126)	10/38	13.5
secondary alcohol biosynthetic process (GO:1902653)	9/34	12.2
regulation of primary metabolic process (GO:0080090)	13/130	11.6
** *TRRUST 2019: Upregulated* **		
PPARA mouse	10/46	11.2
PPARA human	7/39	7.4
PPARG mouse	5/50	4.1
PPARD mouse	3/11	4.0
FOXA2 mouse	4/40	3.4
NR0B2 human	2/8	2.7
** *TRRUST 2019: Downregulated* **		
SREBF1 human	7/27	9.5
SREBF1 mouse	6/36	7.0
SREBF2 human	4/20	5.1
TFAP2A mouse	3/10	4.5
SP1 mouse	9/270	4.2
MLXIPL mouse	3/15	3.9

## Data Availability

Code and data are available at https://github.com/HermanLab/FructoseMetabolismF2Liver (accessed on 1 September 2021).

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
