# Peer review of "A Systems Approach Dissociates Fructose-Induced Liver Triglyceride from Hypertriglyceridemia and Hyperinsulinemia in Male Mice"

_nutrients, 2021, doi:10.3390/nu13103642_

Round 1

Reviewer 1 Report

The present manuscript entitles "A systems approach dissociates fructose-induced steatosis from hypertriglyceridemia and hyperinsulinemia in male mice” try to investigate the dynamic metabolic parameter by high fructose diet in male mice. However, statistical tests and significance should be included in the manuscript should be accurately extended. Below are comments to enhance the manuscript:

  1. Although it is critical for the authors presume that C3H and C57 mice are sensitive and resistant, respectively to fructose-induced hyperinsulinemia and hypertriglyceridemia, hepatic triglyceride in C57 was significantly increased relative to Chow diet. Increasing triglyceride in liver may be release to serum. Is C57 resistant in fructose?
  2. The body weight in C57 induced by fructose was decreased compared to chow diet on end point (HFrD 6week). The author should test to compared to initial point.
  3. All results should be statistically accurately tested. The statics did not appear in figure 2b. In accordance, it should be stated that the differences were statistically different or not statistically different, and p values should be mentioned.
  4.  Although Female F2 cohort is not significant in hyperinsulinemia and hypertriglyceridemia, are the other metabolic traits unchanged in female F2 cohort?
  5. Can steatosis as like fatty liver be caused by fructose induced diet in 4 or 6 weeks? If it so, the author should show histology or Oil red O staining using liver tissue.
  6. Although the author tried to explain which mechanism plays a role in fructose induced diet, F2-cohort did not associate with variant of Srebp1 and tlr4. And they tried to use SCRB-seq for finding transcriptional pathways. Based on SCRB-seq, authors should check the genes found through SCRB-seq.
  7. As known pathway, the author should check serum leptin levels.

Reviewer 2 Report

The relationship between fructose metabolism and fatty liver is an interesting issue. Unlike glucose, high blood glucose causes metabolic diseases, and fructose metabolism in the liver is more related to metabolic diseases. In addition, the tolerance of fructose seems to be related to individual genetic backgrounds. This study uses two mouse strains with different fructose tolerance and the F2 cohort of male mice derived from crossing these two different strains to analyze the relationship between fructose metabolism and fatty liver-related metabolic pathways. The results of this study provide some interesting information, but it is also a little confusing. These questions seem to come from the genetic variation of F2 individuals, which leads to a decrease in the credibility of the results of this study. For example:

  1. From Figure 2, whether it is a metabolite in serum or liver cells, the variation in the F2 cohort is significantly greater than the variation in the parental strains.
  2. Some relevance data in Table 2 are different from ordinary understanding. For example, why is there no significant correlation between high blood glucose and high serum insulin?
  3. Due to the large metabolic variability of the F2 population, the statement that Srebf1 haplotypes are not the main cause of susceptibility to fructose metabolism disorders needs to be further elucidated.
  4. The results of genome and transcriptome analysis do not seem to be very novel.

Round 2

Reviewer 1 Report

The manuscript is much improved.